# Research on Cooperative Communication Strategy and Intelligent Agent Directional Source Grouping Algorithms for Internet of Things

**Yongyan Zou \*, Yanzhi Zhang and Xin Yi**

Electrical & Information Engineering Department, Sichuan Engineering Technical College, Deyang 618000, Sichuan, China; Zhangyanzhi410@163.com (Y.Z.); yixin2018yi@126.com (X.Y.)

**\*** Correspondence: yanyangtian19@scetc.edu.cn; Tel.: +86-1801-6117-068

**Abstract:** In order to improve the network layer of the Internet of things to improve transmission reliability, save time delay and energy consumption, the Internet of things cooperative communication and intelligent agent technology were studied in this paper. In cooperative communication, a new cooperative communication algorithm KCN (k-cooperative node), and a reliability prediction model are proposed. The k value is determined by the end-to-end reliability. After k cooperative nodes are selected, other nodes enter dormancy. In aggregate traffic allocation, game theory is used to model the traffic equilibrium and end-to-end delay optimization scenarios. In practice, the optimal duty cycle can be calculated, which makes some cooperative nodes enter an idle state to save energy. Under the premise of guaranteeing end-to-end delay, it is shown that the reliability of the proposed KCN algorithm is better than that of the other existing routing protocols. In the aspect of intelligent agent, a Directional source grouping based multi-Agent Itinerary Planning (D-MIP) is proposed. This algorithm studies the routing problem of multi-agent parallel access to multiple source nodes. A directed source packet multi-agent routing planning algorithm is proposed. The iterative algorithm of each source node is limited to a sector, and the optimal intelligent agent route is obtained by selecting an appropriate angle. Compared with other algorithms, it is shown through a lot of simulated results that energy consumption and time delay has been saved by the proposed D-MIP algorithm.

**Keywords:** intelligent agent algorithms; internet of things (IoT); cooperative communication; routing protocol; duty cycle; game theory; directional source grouping

---

## 1. Introduction

Cooperative communication refers to selecting multiple forwarding nodes for cooperative communication when forwarding data, which can improve the reliability of transmission; intelligent mobile agent can collect data in the network, and integrate, compress and encrypt data at the same time, which provides additional flexibility and new functions for the Internet of things network layer. Cooperative communication needs to consider various energy saving measures. In high dynamic networks, saving energy while maintaining the reliability of nodes is an important research direction. Although multiple forwarding nodes are used in existing cooperative communication protocols to improve the reliability of nodes. The reliability model is unpredictable. Moreover, using the sleep of the sensing node to save energy is not considered. Therefore, a new cooperative communication algorithm is needed. Multi-agent routing planning is a new research direction of intelligent agent transmission. The multi-agent routing is prone to overlap between routes, which results in energy waste. Furthermore, it is necessary to study new multi-agent routing algorithms.

In recent years, wireless sensor networks (WSNs) are an important part of the Internet of Things (IoT). They have undergone tremendous development [1]. The wireless channel of WSN in the

IoT changes dynamically and the resources of nodes are limited. Thus, it is a great challenge to provide high-performance communication, especially multimedia communication. Cooperative communication is considered a solution to this challenge, which is scalable, energy efficient and fault tolerant [2,3]. The IoT has been applied to various fields and formed intelligence. WSN is the network layer of the IoT. Through intelligent mobile agents, data and deploy tasks can be collected. The optimization algorithms of cooperative communication and intelligent mobile agent are studied. The specific contributions are as follows.

(1) In the aspect of cooperative communication technology, a new cooperative communication algorithm KCN (k-cooperative node) is proposed. K cooperative nodes for transmission are used in each hop. Moreover, the reliability of transmission in dynamic network is improved.

(2) Intelligent mobile agent has a view of the whole network. Sink node centralizes the planning of proxy routing. The proxy routing can be controlled by programming. In fact, the idea of software defining network has been applied in the realization of an intelligent agent. In this paper, the multi-agent routing planning algorithm is studied, and a directional source grouping based multi-agent itinerary planning (D-MIP) is proposed.

The remainder of the paper is structured as follows. Section 2 introduces the current research progress and shortcomings. In Section 3, a new cooperative communication algorithm KCN is proposed. The algorithm uses K cooperative nodes per hop for transmission, which improves the reliability of transmission in dynamic network. Section 4 studies the multi-agent routing planning algorithm and proposes the directed source packet multi-agent routing planning algorithm. Section 5 describes the experimental process and result analysis. Section 6 summarizes the whole paper and puts forward the next research plan.

## 2. Related Research

Sensor nodes in WSNs are composed of three types: Sink node, data source sensor node for sensing information and relay sensor node for forwarding data. In the intelligent agent algorithm of IoT network layer, data is collected by intelligent agent. The intelligent mobile agent is made up of control node, address information and carrying data. It starts from the Sink node; then, the multiple source nodes to collect data are visited. Moreover, the data sensed by the data source sensor are compressed and fused by the intelligent agent. Finally, data are brought back to the Sink node. Compared with the traditional client-server computing model of WSNs, the efficiency of data collection and aggregation based on intelligent agent is higher [4]. In recent years, wireless sensor networks use the concept of mobile agent to reduce energy consumption and obtain effective data collection. Generally, in the data collection based on a mobile agent, it is an important and necessary step to find out the optimal travel planning of a mobile agent. However, with the expansion of network scale, single agent travel planning has two main disadvantages: task delay and large scale of mobile agent. Therefore, multi-agent travel planning overcomes the shortcomings of single agent travel planning [5].

The intelligent agent design of WSNs can be divided into four parts: general framework, route planning, middleware design and agent cooperation. The main algorithm of intelligent mobile agent is to plan the mobile route of agent. Because the Sink node already knows the location of all source nodes, the route planning work is completed in the Sink node. That is, the Sink node has a global view. The route planning of agent is centrally controlled by Sink node, which coincides with the idea of software-defined network.

The disruptive potentials of the IoT entail multifaceted requirements and development issues (large scale deployments, heterogeneity, cyberphysicality, interoperability, distributed smartness, self-management, etc.). To adequately tackle them and to comprehensively support the development of the IoT ecosystem, the Agent-Based Computing (ABC) represents a proper and solid modeling, programming and simulation paradigm [6]. WSNs have many potential applications, such as battlefield surveillance, medical applications, field surveillance and disease prevention. WSNs need

data transmission to be processed. However, due to the dynamic changes of wireless channel in WSN, providing performance communication becomes a major challenge. Because of energy limitation, the lowest energy consumption is required for the sensor nodes on the premise of ensuring reliability. The technology to solve this problem is cooperative communication, which is considered a scalable, energy-saving and fault-tolerant solution. Cooperative communication nodes cooperate with each other to form multiple forwarding paths. The performance of WSN, especially the reliability of the network, can be improved by cooperative communication through utilizing the smart and spatial distribution of wireless media. A multi-mobile agent travel planning algorithm (CL-MIP) is proposed in the literature [7]. It divides the problem of multi-mobile agent WSNs into four steps: (1) calculating the density center of the source node in the network; (2) taking the density center as the center of circle, with the predefined radius, the source nodes that are covered are grouped into one group; (3) computing the order in which each mobile agent accesses the source nodes in the group; (4) continuing to iterate until there is no unmarshalled node in the network, and sending multiple mobile agent packets in parallel. A system structure for the research of multi-mobile agent travel planning is provided by CL-MIP. In the travel planning of multiple mobile agents, the first thing that needs to be considered is the problem of source node grouping in the network. Reasonable and effective marshalling can transform the problem of multiple mobile agents into a parallel problem of multiple single mobile agents. In CL-MIP, the gravity model in physics is imitated to calculate the density of source nodes, and full consideration of the geographic location relationship between source nodes is made. As a basis for grouping, the convergence rate of each mobile agent migration can be effectively improved. However, in the specific marshalling steps, there is still no perfect solution on how to choose the appropriate marshalling radius. Therefore, the effectiveness of grouping the core issues in this multi-mobile agent requires more verification. One research in the literature [8] considers the problem of learning cooperative policies in complex, partially observable domains without explicit communication. This work extends three classes of single-agent deep reinforcement learning algorithms based on policy gradient, temporal-difference error and actor-critic methods to cooperative multi-agent systems. Another research [9] defines group-based topologies and show how some wireless ad hoc sensor networks (WAHSN) routing protocols perform when the nodes are arranged in groups, connections between groups are established as a function of the proximity of the nodes and the neighbor's available capacity (based on the node's energy). The importance of evaluating localization algorithms in real nodes is proven in the literature [10]. Furthermore, it is used to avoid the overhead that resource-constrained nodes may not be able to bear. Mobile agent routing planning problem for single task is analyzed; node energy consumption and time consumption are two main factors in the process of mobile agent migration. The constraint conditions and objective function model of mobile agent routing problem between nodes for single task are established. An improved Dijkstra algorithm was designed to solve this model in the literature [11]. The simulation result shows that the proposed algorithm can get optimal mobile agent route between any two nodes in the wireless sensor network according to the current network environment. Research [12] has proposed that physical layer cooperative communication can significantly improve network capacity. Another researches [13] studied the cooperative communication between Long Term Evolution (LTE) cells, especially multimedia communication. One study [14] proved the importance of evaluating the location algorithm in the actual nodes to prevent the introduction of a large amount of overhead that the resource constrained nodes may not be able to bear.

Single-mobile agent itinerary planning (SIP) and multi-mobile agent itinerary planning (MIP) are included in intelligent agent path planning algorithms. In a single intelligent proxy algorithm, based on the minimum distance of the node Sink. The scalability of the single agent algorithm is poor. The distance between source nodes that is suitable for scenarios is close, and the number of source nodes is small. For large-scale WSNs, many problems occur in single-agent data, such as large latency, unbalanced load and low reliability.

The data in the sensing layer of the Internet of things is diverse. In the wireless multimedia sensor network, the data not only includes text, but also multimedia data such as audio and video data. Because of the limited resources, the sensor nodes cannot upload all the data; thus, it is of great significance to study the energy-saving and intelligent path algorithm. Cooperative communication refers to selecting multiple forwarding nodes for cooperative communication when forwarding data, which can improve the reliability of transmission; an intelligent mobile agent can collect data in the network, and integrate, compress and encrypt data at the same time, which provides additional flexibility and new functions for the Internet of things network layer. Compared with the traditional client server computing model of wireless sensor network, the efficiency of data collection and aggregation based on intelligent agent is higher; the bandwidth and delay requirements of multimedia are higher. It is of great significance to study these two algorithms for improving transmission reliability and saving time delay and energy consumption in the Internet of things network layer.

## 3. Cooperative Communication Algorithms

### 3.1. Cooperative Communication

#### 3.1.1. Beaconless Routing Protocol

In the positioning of WSNs, geographic routing has become an attractive solution due to its scalability and efficiency. Most existing geographic routing is stateful. That is, each node needs to maintain the location information of its immediate neighbors. General Packet Radio Service (GPSR) is the most representative geographic routing to get the location of neighbors by broadcasting beacon information regularly. Based on neighbor information table, the next hop is chosen by stateful routing. In highly dynamic networks, it has the following shortcomings: (1) excessive memory and communication overhead are caused by maintaining neighbor information, which leads to energy consumption; (2) the collected neighbor information is quickly out of date. Moreoever, data packets need to be sent repeatedly, which leads to excessive energy consumption.

In order to overcome the shortcomings of traditional geographic routing, a stateless geographic routing protocol (beacon-free routing protocol) is proposed. In stateless routing, each node does not need to maintain neighbor node information and send beacons regularly. The next hop node is selected according to the competition mechanism. Neighbor nodes start a certain countdown. The node whose countdown first ends has the forwarding right. Therefore, with high reliability, the forwarding nodes are chosen by beacon-free geographic routing according to the actual topology. However, the node countdown time is too close, which may cause a conflict between nodes. Furthermore, the energy efficiency, delay and reliability may be affected. In order to avoid conflicts between potential forwarding nodes, a competitive timer is needed. However, how to set a timer to avoid conflicts and delays is still controversial. To solve this problem, the area where potential forwarding nodes are located is limited in some protocols, such as sector, triangle and circle. Reliable and Energy-Efficient Routing (REER) specifies forwarding nodes through the nodes in a circular overlap area. The two principles of setting forwarding areas are as follows: (1) the nodes in the area can receive messages from neighboring potential forwarders (PE); (2) to reduce or avoid conflicts, the nodes in the region set a dynamic countdown delay.

Nodes in cooperative communication cooperate with each other, and multiple forwarding paths can be formed without multiple antennas. Wireless media has the characteristics of broadcasting and spatial distribution. Thus, the performance of WSN, especially the reliability of islanding network, can be improved by cooperative communication. The cooperative communication transmission mode is shown in Figure 1. Source nodes send data to Sink, and multiple forwarding nodes were selected for each hop to improve reliability.

In most existing cooperative communication research, nodes are accessed through orthogonal channel (FDMA, TDMA, CDMA). Channel is shared among participating nodes (source, target, cooperative node), and only one hop of communication is studied. In this paper, a new multi-hop

cooperative communication algorithm KCN is proposed, and fixed k cooperative nodes per hop are chosen.

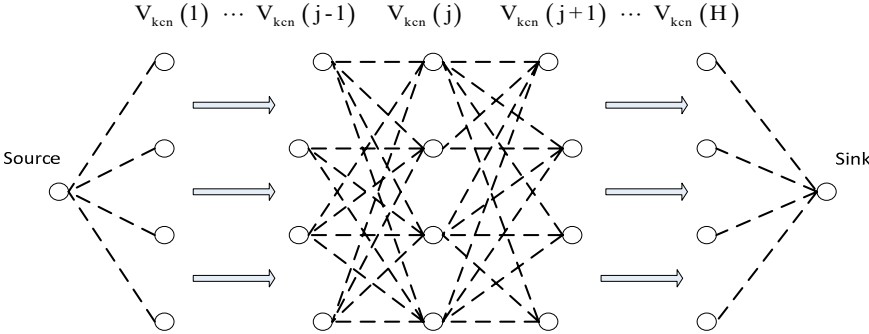

**Figure 1.** Collaborative Node Communication.

### 3.1.2. KCN Algorithm

When choosing the cooperative node, the source node broadcasts and sends PROB messages, and K cooperative nodes in each hop are determined by a certain algorithm. Among all cooperative nodes in each hop, one node is chosen as reference node (RN). When data is forwarded, the reference node assigns a backward delay to each cooperative node, and the skillful points with high performance will be assigned high priority. After receiving the broadcast packet, the backward delay is started by the cooperative node, and the node with the shortest countdown will continue to broadcast the packet. The comparison between KCN (k-cooperative node) and beacon-free routing is shown in Table 1.

**Table 1.** Comparison of KCN and beacon-free routing.

|  | Beaconless Geographic Routing | KCN |
|---|---|---|
| Number of potential forwarding nodes per hop | The number is uncertain and the reliability cannot be guaranteed | The quantity is k, and the reliability can be predicted. |
| Delay of forwarding data | Three-time handshake is required, and the time is extended and uncertain. | Reference nodes are selected centrally, allocated slots are set, and the delay is smaller. |
| Remaining Energy of Forwarding Node | Not considered | Will consider |
| Regular redistribution | No | Yes |
| Node dormancy | No | Yes |

In KCN, under the premise of ensuring the reliability, the number of activated cooperative nodes is reduced to optimize network performance, and an ETE reliability model is established to explore the relationship between several parameters. These parameters include duty cycle, link failure rate, the average node density, the number of cooperative nodes in one hop, the hop count, and the reliability, so that the optimal parameters are set.

### 3.2. End-to-End Reliability Model

Given duty cycle $\mu$ and cooperative node $K$, the probability formula for calculating the number of active cooperative nodes $k$ is as follows:

$$P_k = C_K^k \times \mu^{K-k} \times (1-\mu)^k \tag{1}$$

Given $k$ active cooperative nodes, the probability of successful transmission of the current hop is $W_k$:

$$W_k = P_k \times \left(1-f^k\right) = C_K^k \times \mu^{K-k} \times (1-\mu)^k \times \left(1-f^k\right) \tag{2}$$

Assuming that the number of active cooperative nodes in hop $j$ is $k$, the sending failure rate of node $k$ is $f_k(j)$:

$$f_k(j) = P_k \times f^k = C_K^k \times \mu^{K-k} \times (1-\mu)^k \times f^k \tag{3}$$

Then, the failure rate of hop $j$ is the sum of the failure rate of each node:

$$f_k(j) = \sum_{k=1}^{k=K} f_k(j) = f_1(j) + f_2(j) + \cdots + f_K(j) \tag{4}$$

The successful transmission rate of pick $j$ is $W(j)$:

$$
\begin{aligned}
W(j) &= 1 - f(j) \\
&= 1 - f_1(j) - f_2(j) \ldots - f_K(j) \\
&= 1 - C_K^0 \times \mu^K - C_K^1 \times \mu^{K-1} - C_K^2 \times \mu^{K-2}\left(1 - \mu^2\right)f^2 \\
&\quad - \ldots - C_K^{K-1}(1-\mu)^{K-1}f^{K-1} - C_K^K(1-\mu)^K f^K
\end{aligned}
\tag{5}
$$

$C_K^0 \times \mu^K$ means that all KCNs fail to sleep, and $C_K^1 \times \mu^{K-1}$ means that one node is active, but fails to send. $C_K^2 \times \mu^{K-2}\left(1 - \mu^2\right)f^2$ denotes that this jump has two active nodes, but all send failed. By analogy, $C_K^K(1-\mu)^K f^K$ means that all $K$ nodes are active but fail to send.

Therefore, end-to-end reliability $W = W(1) \times \ldots \times W(j) \times \ldots \times W(H)$. It is assumed that the reliability of each hop is the same.

$$W = W(j)^H \tag{6}$$

For network users, it is considered that they prefer as little end-to-end delay as possible to get network services faster. It is assumed in this paper that the delay is the sum of the queuing delays of the links in the transmission path. According to the M/M/1 model, the queuing delay of link $(i, j) \in E$ is defined as:

$$\tau_{i,j} = \frac{1}{c_{i,j} - u_{i,j}} \tag{7}$$

The average queuing delay $\tau_p^{s,d}$ of path $p \in P^{s,d}$ is the sum of queuing delays of each hop, that is:

$$\tau_p^{s,d} = \sum_{(i,j) \in E} \frac{\delta_{i,j}^{s,d}(p)}{c_{i,j} - u_{i,j}} \tag{8}$$

When link $(i, j)$ is on path $p$, $\delta_{i,j}^{s,d}(p)$ is 1; otherwise, it is 0, and the end-to-end delay between source-destination $(s, d)$ is:

$$\tau^{s,d} = \sum_{p \in P^{s,d}} \alpha_p^{s,d} \tau^{s,d} \tag{9}$$

According to the network users' requirement of minimizing the expected delay, the average end-to-end delay of user $s$ is defined as:

$$\frac{1}{t(s)} \tag{10}$$

Assuming that there are multiple users in the network, each user wants to get the shortest average end-to-end delay. Obviously, due to the overlap of end-to-end paths, one user may be affected by the delay of another user. As a participant in the game, the end-to-end delay is the only income indicator. In this paper, the user's income function is defined as:

$$\prod_s \frac{1}{t(s)} \tag{11}$$

The distribution of traffic on each path will be adjusted by the reasonable users to maximize their income function. In order to achieve this goal, users in the game will cooperate with each other to achieve a "win-win" scenario. In this case, the product of collective income function is maximized.

According to the above model, joint optimization is used to model for the providers and users of the network service. The purpose of optimization is to calculate a traffic configuration $\alpha_p^{s,d}$ to make the network link load relatively balanced. At the same time, users can experience shorter delay [15,16]. It can be described with the following optimization model:

$$\max_{|\alpha_p^{s,d}|}\left(\Pi_p\left(1-\theta_p\right) + \Pi_s \frac{1}{t(s)}\right) \tag{12}$$

$$\sum_{(s,d)}\sum_p h_{derive}^{s,d}\cdot\alpha_p^{s,d}\delta_{i,j}^{s,d}(p) \le c_{i,j} \quad \forall(i,j) \in E \tag{13}$$

$$\sum_p \alpha_p^{s,d} = 1 \quad \forall s,d \in V, s \neq d \tag{14}$$

$$\alpha_p^{s,d} \ge 0 \quad \forall s,d \in V, s \neq d, \forall p \in P^{s,d} \tag{15}$$

Formula (14) ensures that the traffic on the link is not larger than the link capacity to avoid congestion. Formula (15) ensures that all requirements are transmitted over the network. Formula (14) has $M$ constraints, Formula (15) has $N^2 - N$ constraints and Formula (16) has $\left(N^2 - N\right)|P_{avg}|$ constraints. Among them, $|P_{avg}|$ is the average number of paths between nodes and $O\left(N^2|P_{avg}|\right)$ is the number of variables.

### 3.3. Model Validation

According to the formulas of reliability model, the effects of $K$, $\mu$, $f$ and hops on $W$ are shown below.

3.3.1. The Effect of Hops and Number of Cooperative Nodes on End-to-End Reliability

$K$ is set from 1 to 10, the number of hops varies from 4 to 16 and the step size is 4. The result is shown in Figure 2. The number of cooperative nodes leads to the increase of reliability, and the increase of hops leads to the decrease of reliability.

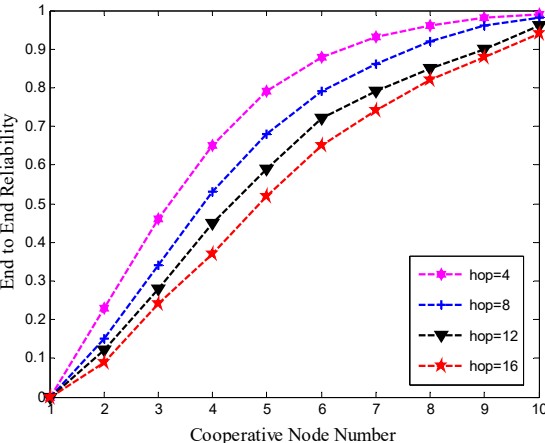

**Figure 2.** The effect of hops and number of cooperative nodes on end-to-end reliability.

### 3.3.2. The Effect of $f$ and $k$ on End-to-End Reliability

B is set from 1 to 10, $f$ from 0.15 to 0.75, step size is 0.15, and $\mu$ is fixed to 0. In order to keep up with the simulation scenario, the number of hops is set to 14. It is shown in the simulation results that the lower $f$, the larger $K$ and the higher end-to-end reliability. In order to increase the success rate of sending packets, Forward Error Correction (FEC) coding is used. The FEC-coded packets can recover the whole package as long as more than two-thirds of the packets are received. Therefore, under a given $f$, the minimum $K$ value can be found according to Figure 3a. The pink region represents more than 2/3 reliability. If $f = 0.3$, then $K$ will be greater than 3. If $f = 0.45$, then the minimum value of $K$ will be increased to 5.

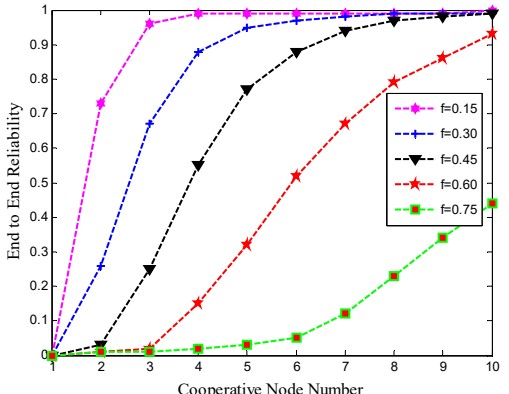

(**a**) Effect of f and k on the End-to-End (ETE) Reliability

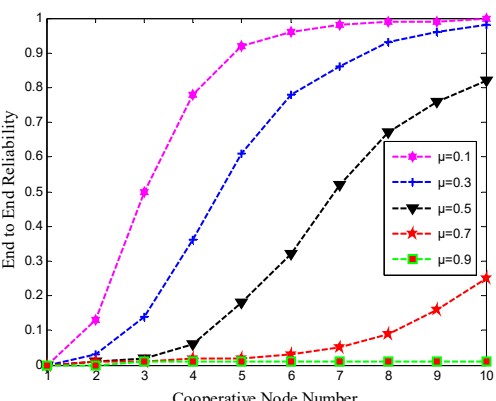

(**b**) Effect of f and $\mu$ on the End-to-End (ETE) Reliability

**Figure 3.** f and k effect on end-to-end reliability.

### 3.3.3. The Effect of $f$ and $\mu$ on End-to-End Reliability

In Figure 3b, $K$ varies from 1 to 10, $\mu$ varies from 0.1 to 0.9, $f$ is set to 0.3, hops are still set to 14. It is shown that the smaller $\mu$, the larger $K$ and the higher reliability $W$. If FEC coding is applied, when $\mu$ is set to 0.1, $K$ should be greater than 4. When $\mu$ is 0.5, $K$ should be set to 9.

### 3.4. Algorithm Design

In the model, the sending and receiving radius of all nodes $(0, R_{\max}]$ is the same. $R_{\max}$ is the maximum transmission range, and each node knows its own location and the location of Sink node. Network topology can be dynamically changed (resulted from nodes periodically dormant, link or node failure). The symbolic description in the model is shown in Table 2.

Graphs $G = (V, E)$, $V = \{v_1, v_2, \ldots, v_N\}$ and $N = |V|$ are used to represent the number of nodes by multi-hop WSN nodes. When $|uv| \leq R$, $|uv|$ refers to the linear distance between node $\mu$ and $v$. Next, the KCN algorithm is described in two steps.

A. Selecting Cooperative Nodes

Given the reference node $m$, then $C_m = \{C_m^1, C_m^2, \ldots, C_m^{k-1}\}$ represents $k - 1$ cooperative nodes of $m$, and $R_{KCN}$ is selected as the overlap of the following regions by the next hop cooperative node.

The overlapping area of the circle of nodes $C_m^1, C_m^2, \ldots, C_m^{k-1}$, with radius $R$.

1. The circle between $m$ and $f_m$, with radius R/2.

2. Taking Sink as the center of the circle, with radius $r(m)$, $|mt| - R < r(m) < |mt|$.

$V_{kcn}$ refers to the set of KCNs, $V_{kcn} = \{m, C_m^1, C_m^2, \ldots, C_m^{k-1}\}$. For consistency, $m$ can be regarded as $C_m^0$.

**Table 2.** Symbol Description Table.

| Symbol | Explain | Symbol | Explain |
|---|---|---|---|
| $s$ | Source node | $P_k$ | Probability of k awakening at K nodes per hop |
| $t$ | Sink node | $C_m^i$ | Reference node of node m of hop i |
| $H$ | Number of hops from source to target | $R_{KCN}$ | Search area of collaboration node |
| $f$ | Link Failure Rate | $R_{mt}$ | With sink node t as the center of the circle with radius $r_t(m)$, the radius meets formula $|mt| - R < r_t(m) < |mt|$ |
| $\delta$ | Node density | $R_{mf}$ | The diameter of the circle is connected between the ideal position $f_m$ of the next hop of node m. |
| $R$ | Transmission radius | $n_m$ | Number of all neighbors of node m |
| $|uv|$ | Distance between nodes $u$ and $v$ | $v_{kcn}(i)$ | All the cooperative nodes of the i-hop $\{C_m^0, C_m^1, C_m^2, \cdots, C_m^{k-1}\}$ |
| $K$ | Total number of cooperative nodes per hop design | $L_{QE}$ | Residual energy of the node |
| $k$ | Number of active nodes per hop | $W_k(i)$ | The success rate of the i-hop k available collaboration nodes |
| $T_{epoch}$ | Construction cycle of KCN | $W$ | End-to-end reliability |
| $\mu$ | Node duty cycle | | |

In KCN, the selection of reference and cooperative nodes is based on location and residual energy, as shown in Figure 4. When the PROB package of the upstream node is received by node $m$, the ideal location for the next hop is included. The operation of node $m$ is as follows:

(a) In the search domain of cooperative nodes, $k$ cooperative nodes are selected according to distance and energy. The four cooperative nodes in Figure 4 are 1–4, and the formula based on distance and energy is as follows:

$$Q(m) = \sqrt{\left(1 - \frac{d}{R}\right) + \left(\frac{e_{res}}{e_{init}}\right)^2} \tag{16}$$

Formula $d$ is the distance from the cooperative node to the ideal position $f_m$ of the next hop; $R$ is the transmission radius of the node; $e_{res}$ is the residual energy; $e_{init}$ is the initial energy. $K$ nodes are sorted into array by node $m$ in order of $Q$ value from big to small.

(b) The location and residual energy of nodes 1–4 are notified to other cooperative nodes $C_m^1$ and $C_m^2$ of this hop.

(c) The location and residual energy of nodes 1–4 are notified to nodes 1–4.

(d) The cooperative node with the largest $Q(m)$ value is selected. Its corresponding ideal location for the next hop is $f_{m+1}$. The location is put into the PROB message and sent.

The unselected nodes enter a dormant state to save energy consumption. Before the next cooperative node selection, they wake up to participate in the selection.

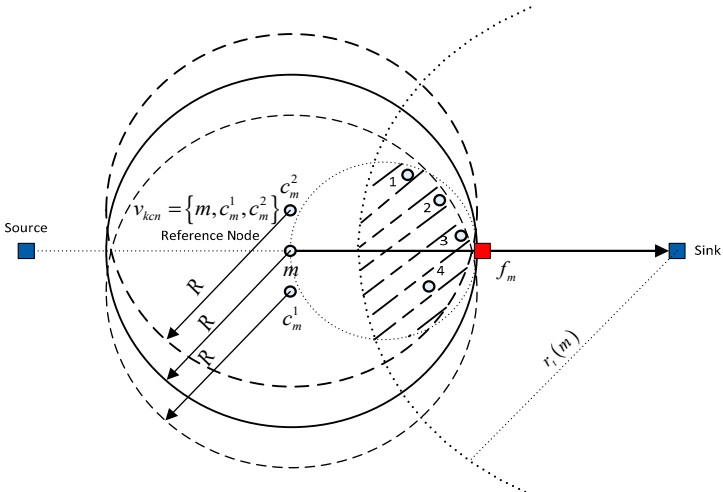

**Figure 4.** Selection of collaboration nodes.

B. Forwarding Packets

(a) Time slot of cooperative node

Assuming that there are $k$ cooperative nodes, $k$ slots are needed to allocate the delay. The $Q$ value of the cooperative node is put into the array from large to small, so that the $Q$ value of the first node in the array is the largest. The waiting time is $\tau$, and the minimum waiting time of the $Q$ value of the $k$ node is $k * \tau$.

(b) Forwarding Packets

The node whose slot expires first forwards the data packet, and other nodes of this hop are notified to cancel the data packet.

(c) PROB transmission cycle

After forwarding data packets for a period of time, the energy of the nodes is decreased. Continuous forwarding will lead to the exhaustion of the energy of the nodes, and a batch of cooperative nodes needs to be replaced. A KCN selection period T is set. When T ends, PROB messages are sent by node to find new $K$ nodes. It is assumed that the period between two PROB messages is T_PROB. Application layer data packets are sent regularly, and the interval between sending data is T_DATA. The relationship between them is shown in Figure 5. T_DATA cycle is taken as a basic cycle. One Idle is inserted before and after PROB to distinguish DATA and PROB packages.

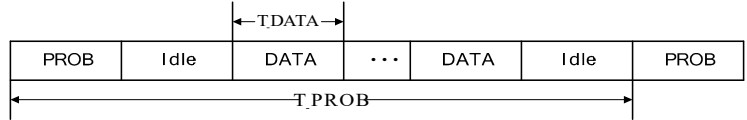

**Figure 5.** The diagram of T_DATA and T_PROB.

Firstly, a parameter is determined; that is, a PROB packet is sent when a $x$ DATA packet is sent. Thus:

$$T\_PROB = T\_DATA * (x + 3) \tag{17}$$

The PROB and DATA are uniformly numbered. It is assumed that the PROB's serial number is S, and $S + x + 3$ is the serial number of the next PROB package. Cooperative node saves the PROB serial number S received by itself, and the DATA packet serial number received is assumed to be $D$. If $D - S \leq x$, it can continue to forward the data packet. If $D - S \leq x$, it indicates that the cooperative node has expired and become a normal node.

### 3.5. Implementation of KCN Algorithm

1. The selection of multiple cooperative nodes of node $m$ is shown in Algorithm 1.
2. The algorithm flow of selecting all cooperative nodes is shown in Figure 6.

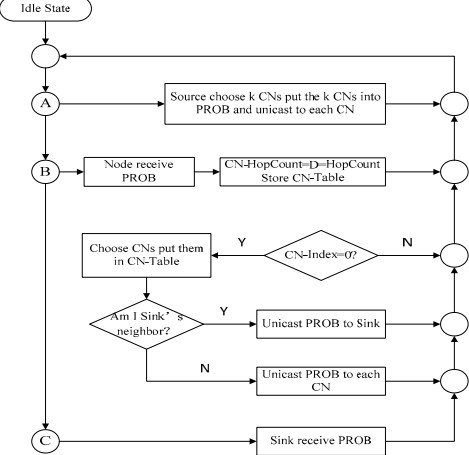

**Figure 6.** Algorithmic flow chart of multi-hop cooperative nodes.

---

**Algorithm 1.** Choose CN ($n_m$,$v_m$,$t$,$R_{KCN}$,$Q$).

---

**Input:** $n_m$ denotes the total number of neighbor nodes of *m*;
　　　$v_m$ denotes the array of neighbor nodes of *m*;
　　　$t$ is the Sink node;
　　　$R_{KCN}$ denotes the cooperative nodes search region of *m*;
　　　$Q$ is array of priority for every neighbor node of *m*;
**Output:** $C_m$ ($C_m$ is array of cooperative nodes of *m*;)
1. For $i = 0$ to K-1 do
2. 　　$Q_{Max} \leftarrow 0$;
3. 　　$idx \leftarrow -1$;
4. 　for $j = 1$ to $n_m$ do
5. 　　　if $V_m[j]$ is $t$ then
6. 　　　　break;
7. 　　　end if
8. 　　　if ($V_m[j]$ in $C_m$) or ($V_m[j]$ not in $R_{KCN}$) then
9. 　　　　continue;
10. 　　end if
11. 　　if $Q[j] > Q_{Max}$ then
12. 　　　$Q_{Max} \leftarrow Q[j]$;
13. 　　　$idx \leftarrow j$;
14. 　　end if
15. 　end for
16. 　$C_m^i \leftarrow V_m[idx]$;
17. 　if $C_m^i$ is $t$ then
18. 　　break;
19. 　end if
20. 　return $C_m$

---

### 3.6. OPNET Simulation

KCN was compared with three other routing algorithms (Beacon-less routing (BLR) [17], REER [18], GPSR [10]). The networks range was set to 1000 m × 500 m, and 600 nodes are deployed randomly. The transmission radius was set to 75 m.

A. Effects of *f* and *k* on *W*

*K* is numbered from 1 to 10, and *f* varies from 0.15 to 0.75. The simulation results are shown in Figure 7; the simulation results of Figure 7a and MATLAB are the same as those of Figure 3a. In Figure 7a, the higher the link failure rate, the larger the number of node retransmissions and the greater the end-to-end delay.

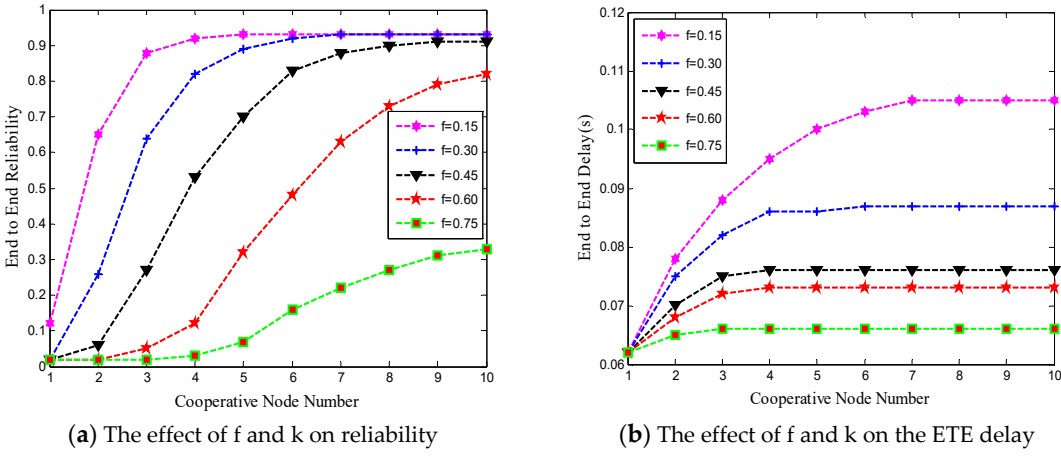

(**a**) The effect of f and k on reliability　　　　(**b**) The effect of f and k on the ETE delay

**Figure 7.** Effect of f and k on end-to-end reliability.

B. The Effect of *f* and *μ* on Reliability

The link failure rate is fixed at 0.3. As shown in Figure 8, *K* varies from 1 to 10, *μ* from 0.1 to 0.9, and step size is 0.2. The simulation results of Figures 3, 4, 5, 6, 7 and 8a and Matlab are similar to those of Figure 3b. The greater *μ* is, the greater possibility of node dormancy is, and the lower reliability *W* is. When *μ* decreases to a certain extent (>= 0.7), the reliability is so low that the packet cannot reach the Sink node. Thus, the delay cannot be counted, and the phenomenon of delay reduction occurs.

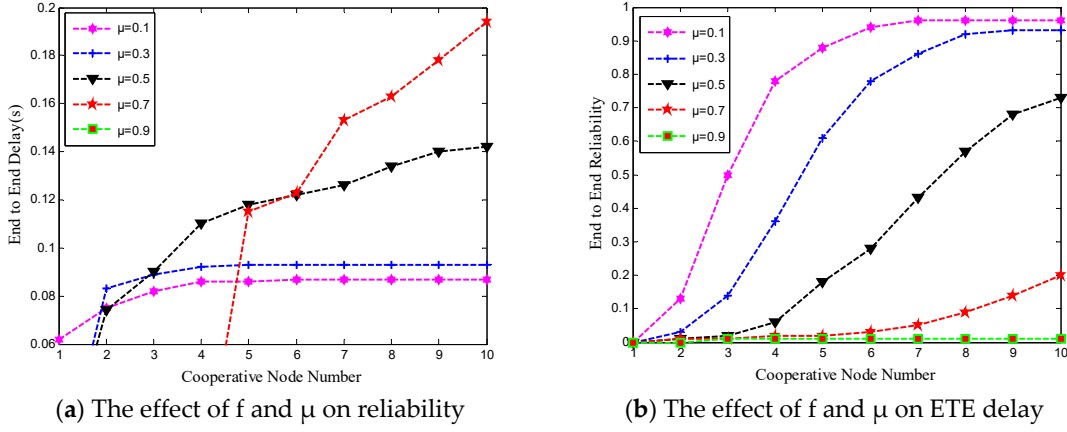

(**a**) The effect of f and μ on reliability      (**b**) The effect of f and μ on ETE delay

**Figure 8.** The effect of f and μ on W.

C. Performance comparison of four routing schemes

The reliability and energy consumption of KCN are compared with three other routing algorithms: REER, BLR and GPSR. The results are shown in Figure 9a. With the increase of link failure rate *f*, the reliability of GPSR decreases rapidly due to the lack of packet loss processing mechanism. BLR periodically re-select relay nodes. Two forwarding mechanisms were used, namely, relay node forwarding and normal forwarding. Broadcasting was used in relay node selection to achieve high reliability. However, once the relay node is selected, it enters the normal forwarding. Under the normal forwarding condition, the relay node cannot process the forwarding according to the channel condition. Thus, its reliability also decreased as quickly as GPSR. In simulation, the same three-step handshake protocol (DATA, REP, SEL) was used for BLR and REER.

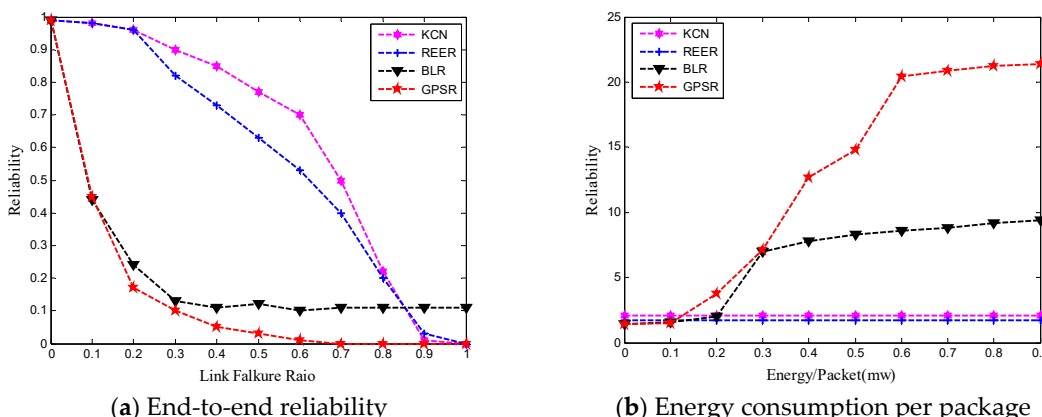

(**a**) End-to-end reliability      (**b**) Energy consumption per package

**Figure 9.** Comparisons of Four Routing Algorithms.

The energy consumption per packet of the four algorithms is shown in Figure 9b. When the link failure rate is greater than 0.06, the energy consumption per packet of GPSR and BLR is decreased to 0. Thus, the energy consumption is no longer counted. BLR can cancel out some link failures when broadcasting, and its energy consumption per packet is low. In KCN and REER, non-cooperative nodes enter dormancy in the data transmission phase. Thus, their average energy consumption per

packet is low. With the increase of link failure rate, the reliability of REER is 0. KCN can maintain high reliability because of fixed *K*-node forwarding nodes. Meanwhile, the hop count is higher than that of REER, and it results in a slightly higher energy consumption per package than that of REER.

## 4. Intelligent Agent Algorithms

Intelligent agent algorithm is actually a software-defined routing algorithm. Its control plane and forwarding plane are separated. The control plane is in the Sink node. The forwarding plane is multiple source nodes, and the intelligent agent is controlled by centralized programming [19,20].

In this chapter, the parallel access of intelligent multi-mobile agents to multiple source nodes for routing planning is studied. The iteration algorithm of each source node is limited to the interior of a sector. The length of the route is controlled by the angle of the directional sector. The optimal intelligent proxy line can be obtained by choosing the appropriate angle.

### 4.1. Multiple Intelligent Agent Route Planning (MIP)

The design of intelligent agent in WSNs can be divided into four parts: general framework, route planning, middleware design and agent cooperation [21]. The access order of source nodes in MA mobile process is determined by the route planning in these components. The key is to find out the optimal route of MA to access a given set of source nodes; it has been proven to be a hard problem for NP.

The sequence of nodes accessed by agents in the mobile process is determined by Single Agent Routing Planning (SIP). The heuristic SIP algorithm [22] and genetic algorithm [23] are usually used to calculate suboptimal solutions. IEMF/IEMA performs better in energy efficiency and delay than other existing algorithms [24,25]. However, when a large number of source nodes need to be accessed, the scalability of the single agent algorithm is not good. SIP is suitable for cases where the distance between source nodes is close and the number of source nodes is small. For large-scale WSNs, many nodes need access. The following problems are presented in single agent data.

① Larger latency: when a single agent needs to access hundreds of nodes.

② Load imbalance: the use of a single agent results in two kinds of energy consumption imbalance. For the whole WSN, all loads are based on a single stream. Nodes in the proxy route will consume more energy than other nodes. When the proxy collects data along the node, the size of the data carried by the proxy continues to increase, and the energy required to forward increases.

③ Reliability decreases: the probability of data loss increases with the continuous increase of data in the process of agent data movement (due to noise and interference in wireless media). Therefore, the longer the route, the lower the reliability.

### 4.2. Directional Source Packets for Multi-Agent Routing Planning

A multi-agent algorithm D-MIP for oriented source grouping is proposed. The number of MAs to be used and the route that MAs need to go through is statically determined by Sink node. Compared with previous studies, the main contribution lies in the introduction of directed source grouping algorithm. It divides the network area into multiple sectors. And the center of the sector is the direct neighbor of Sink node. The basic principle is that, after each MA leaves Sink, it extends naturally and covers the whole area. The typical route covers one sector.

#### 4.2.1. Directional Source Grouping Algorithms

As shown in Figure 10, PE area refers to the circular with radius $\alpha R_{\max}$ and the center of PE, $R_{\max}$ refers to the maximum transmission range (0,1). *K* refers to the number of neighbors of the source node in PE area. These nodes may become the starting nodes of each MA movement, as shown in Figure 10. When $K = 4$, up to four routes can be created by algorithm. Through controlling $\alpha$ parameter, the size of PE region can be adjusted, so that the number of MA can be adjusted.

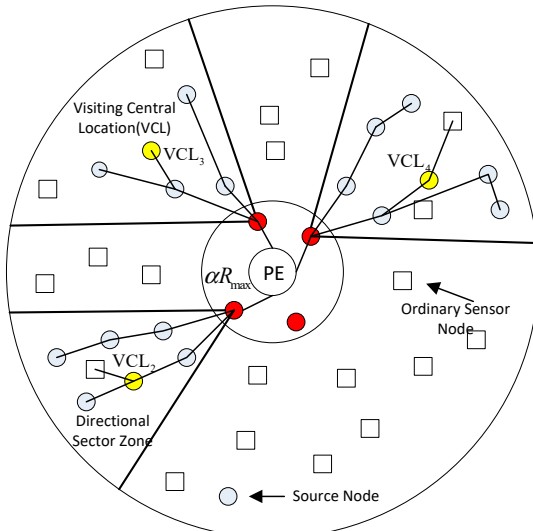

**Figure 10.** Explanation of Directional Source Grouping Algorithms.

Algorithm 2 is an algorithmic pseudocode for oriented source grouping. Firstly, the source node grouping (such as T) is initialized. $Z_{PE}$ refers to the direct neighbor of Sink node in PE area, with radius $SN_j$ (within $R_{max}$). As the algorithm iterates, the number of available start nodes decreases; $Z_{PE}'$ refers to the remaining start nodes. If $|Z_{PE}'| = 1$, all remaining nodes are selected into T. In Figures 3–12, VCL$_2$ is selected to generate the second proxy route. And the node closest to VCL$_2$ line in PE (such as node $e$) is selected as the initial node. Secondly, the center line of the directional sector can be obtained by connecting VCL$_2$ and $e$. As is shown in Figure 10, given the angle $\theta$, the directional sector can be obtained.

---

**Algorithm 2.** Directional source-grouping(V,CL).

---

**Input**: V is the set of source nodes, CL is the central location of source nodes
**Output**: T denotes the set of source nodes to be grouped, SN represents sensor node j within $Z_{pe}$
**Notation:** V′ is the set of remaining sources, part of which will be allocated to an MA.

    n is the total number of sources.
    α is a constant, which is set to 70% in this paper.
    $R_{Max}$ is the maximum transmission range of the sensor node.
    $Z_{pe}$ is the PE-zone which includes the sink node and its one hop neighbors within a radius of $αR_{Max}$.
    $SN_j$ represents sensor node j within $Z_{pe}$.
    K is the maximum number of the starting points for MA itineraries, K $= |Z_{pe}|$.
    $Z_{pe} = \{SN_j: d(PE, SN_j) \le αR_{Max}, j = 1, \dots, K\}$.
    $Z_{pe}$ is the set of remaining starting points, one of which will be allocated to an MA.
    $Z_j$ is the j-th directional sector zone for source-grouping.
    θ is the selection angle of a directional sector zone.

1.     T ← {};
2.     if $|Z'_{pe} = 1|$ then
3.        T ← V′;
4.     else
5.       for each $SN_j$ in $Z'_{pe}$ do
6.          set line(VCL,PE) as the line connection PE and CL;;
7.          calculate the distance d($SN_j$, line(VCL,PE) between $SN_j$ and line(VCL,PE);
8.          if d ($SN_j$, line(VCL,PE)) = min{d($SN_k$, line (VCL,PE)) | $SN_k ∈ Z'_{pe}$ } then
9.            select SN←$SN_j$ as the starting point;
10.            break;

---

---

**Algorithm 2.** *Cont.*

---

11.          end if
12.       end for
13.       set line(SN$_j$,VCL) as line connecting
14.       partition Z$_j$ with line(SN$_j$, VCL) as the central line, and θ as the selection angle;
15.       for each source v in V' do
16.          if v ∈ Z$_j$ then
17.             T ← T ∪ {v};
18.          end if
19.       end for
20.    end if
21.    return T and SN;

---

### 4.2.2. Iterative Algorithm

In each iteration, a new source node group (CL) is established for the remaining source nodes; then, a series of source nodes are assigned for the new intelligent agent. The single proxy algorithm can be used. According to the source nodes grouping obtained in Section 4.2.1, single proxy algorithm can be based on loop traversal or tree traversal. If there are active nodes, repeat the above process until all source nodes are assigned to intelligent agents. Algorithm 3 is the pseudocode of the iterative multi-agent algorithm. In Figure 10, u and V are still left in the nodes after three iterations. In this case, after several iterations, if there are still isolated nodes, u and V are simply assigned to the starting node F of the last iteration.

---

**Algorithm 3. MIP(V)**

---

**Input:** V is the set of original sources to be visited
**Output:** Iterations of MAs 1,2, . . . k).
**Notation:** V' denotes the set of remaining source nodes, currently.
          T denotes the set of grouped sources at each iteration.
          k denotes the index of MAs.
1.  V ' ← V;
2.  k ← 0;
3.  loop;
4.       if V ' ≠ 0 then
5.          k ← k = 1;
6.          CL-CL-selection(V');
7.          (T, SN) ← source-grouping(V', CL);
8.          V ' ← V − T;
9.          determine (Itinerary of MA k) to visit sources T with SN as starting point by some SIP algorithm;
10.       end if
11. end loop
12. return (Itineraries of MAs 1,2,...,k);

---

### 4.2.3. Sector Angle Designation

In SG-MIP, radius is an important parameter in source node grouping operation. In D-MIP, it is solved that how to determine the sector midline. However, how to choose the right angle $\theta$ is still a challenge. The size of the sector is also determined by sector angle $\theta$. Firstly, $\theta$ can be selected based on WSN node density. Meanwhile, $\theta$ should be changed dynamically according to different routes. As shown in Figure 10, some sectors need a large angle such as VCL$_1$ to contain all source nodes in the corresponding direction. Finally, after several rounds of route planning, fine-tuning can be used to

handle isolated nodes. For example, simple grouping can be used to merge the remaining isolated source nodes into the last route. The algorithm should be able to add source nodes one by one, and plan the route according to the shortest path to minimize the added cost.

## 5. Simulation Verification

### 5.1. Simulation Settings

The networks model of document is used [7]. Eight hundred nodes are unevenly distributed in the range of $1000 \times 500$ m. Sink nodes are in the center of the scene, and multiple source nodes are randomly distributed in the network. The parameters of MA system are shown in Table 3; the D-MIP algorithm, E-MIP algorithm [26] and Dijkstra algorithm [11] are compared.

**Table 3.** Setting of simulation parameters.

| Parameter | Value | Parameter | Value |
|---|---|---|---|
| Raw data size | 2048 bits | Network size | $1000 \times 500$ m |
| MA code size | 1024 bits | Radio transmission range | 60 m |
| MA accessing delay | 10 ms | Number of sensor node | 800 |
| Data processing rate | 100 Mbps | Media Access Control (MAC) layer standard | 802.11b |
| Aggregation ratio | 0.8 | | |

### 5.2. Evaluation Criterion

In order to evaluate the algorithm, the following three evaluation criteria are proposed:

A. Energy consumption: in the simulation, the energy consumption of all sensor nodes in mobile agent task processing and wireless communication is recorded. This is then taken as an evaluation criterion of whether the algorithm can effectively save network energy.

B. Task Delay: The time when the first mobile agent is sent to the network to the time when the last mobile agent returns to the data center is recorded as the task delay. Task delay is an important parameter to evaluate the performance of the algorithm.

C. Energy Delay Product (EDP): A flexible response to different application requirements through the control of parameter $\alpha$ is an important feature of this algorithm. However, for the sake of quantitative evaluation, it is necessary to set up a specific application. In some practical applications, the effectiveness of energy and the delay of tasks in the network need to be guaranteed; this requires us to take both of them into account. EDP calculates the product of energy and task delay, and it takes the performance of both energy and delay into account. Therefore, EDP can be used as a criterion to evaluate the overall performance of the algorithm.

### 5.3. Measurement and Selection of Control Parameter $\alpha$

In Section 4.2.1, the directional source grouping algorithm can adjust the size of the PE region by controlling parameters, and it can adjust the number of MA. Therefore, in the simulation, the influence of the $\alpha$ value on the performance of the proposed algorithm should be examined first. A scenario of 15 source nodes is set up, and a large number of random seeds are used to simulate the impact of energy, delay and the influence of EDP with the change of $\alpha$.

As can be seen from Figure 11a,b, when $\alpha$ is less than 0.5, the energy consumption in the network remains at a high level, while the task delay is the smallest. This is because the distance between the source node and the data center occupies an excessive proportion in the weight calculation of undirected connected graphs. This extreme condition results in all source nodes to be connected to the data center in the construction of the minimum spanning tree. That is, after the multi-mobile agent marshalling, a single mobile agent is assigned for each source node by the data center. Obviously, task latency can be minimized at this time. However, due to the fact that the advantages of data fusion of

mobile agents have not been fully exploited, the energy consumption of mobile agents is maintained at a high level. It is worth noting that when the control parameter $\alpha$ is greater than 0.5, the energy consumption in the network is decreased gradually with the increase of trade-off factor, while the delay of tasks is increased gradually. This trend tends to flatten when the control parameter $\alpha$ approaches 1.0. The curve trend in the graph fully illustrates the inverse relationship between energy consumption and task delay. Thus, the required multi-mobile agent travel planning algorithm is obtained by adjusting the control parameter $\alpha$.

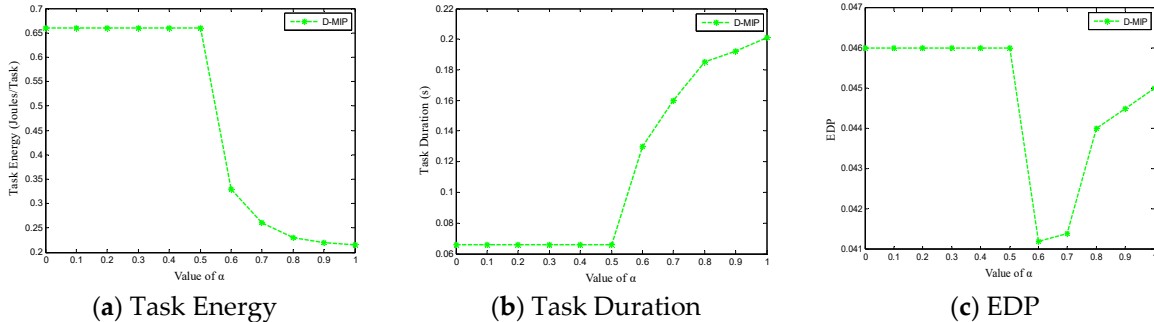

(**a**) Task Energy            (**b**) Task Duration            (**c**) EDP

**Figure 11.** Corresponding Relationship between Energy Consumption, Task Delay, EDP and Control Parameter α.

In the last section, energy delay product EDP is proposed to be taken as a comprehensive performance index. Therefore, the corresponding relationship between EDP and control parameter $\alpha$ in the experiment need to be examined. Obviously, the relationship between energy consumption and task delay is increasing and decreasing respectively. There must be an optimal solution for the EDP value, which is the reflection of their comprehensive performance. From the curve of Figure 11c, it can be seen that, when the value of control parameter $\alpha$ is less than 0.5, the performance of EDP remains unchanged; this is directly related to the stability of energy consumption and task delay in this range. When the control parameters $\alpha$ continue to increase to 0.6, the EDP values continue to decrease. This means that the comprehensive performance of the algorithm has been continuously improved. However, when the control parameter $\alpha$ continues to increase to 1.0, the EDP value increases; this means a drop in the performance of the algorithm. Therefore, when the control parameter $\alpha$ is 0.6, the proposed algorithm can achieve the best comprehensive performance of EDP.

*5.4. Performance Comparison and Analysis*

In this section, the effect of the number of source nodes on energy consumption and delay is examined. The number of source nodes was set to vary from 10 to 40, with a step size of 5. The effect of the number of source nodes on energy (the sum of all the energy transmitted, received, crosstalk and collision in the process of data acquisition by source nodes) is shown in Figure 12. It can be seen that the energy consumption of the three algorithms increased with the increase of the number of source nodes. When the number of source nodes increases from 10 to 40, the energy of E-MIP changes from 0.2 J/task to 0.54 J/task. By contrast, Dijkstra is about 0.07 J less than E-MIP, while D-MIP is about 0.02 J less than Dijkstra.

The comparison of task duration of the three algorithms is shown in Figure 13. The task duration of E-MIP algorithm is from the time MA, leaving the Sink node to the time agent returning Sink. This is the same as the average end-to-end delay. In the MIP algorithm, multi-agent tasks are executed in parallel, and the duration of tasks is the delay of agents, so that the number of source nodes has a greater influence on the delays than on energy consumption. In the E-MIP algorithm, as the number of source nodes increases, the delay increases rapidly. As the number of source nodes (n) increases, the agent carries more and more data. Compared with E-MIP, in the Dijkstra algorithm, when n = 40, the task duration is less than 0.5 s. The reason is that all source node data distributed in the whole network

need to be collected by a single agent, which causes a large delay. Multi-agent concurrent execution speeds up the task process. The delay of D-MIP is the lowest among all algorithms. The delay of D-MIP algorithm is 0.1 s lower than Dijkstra, when n = 40.

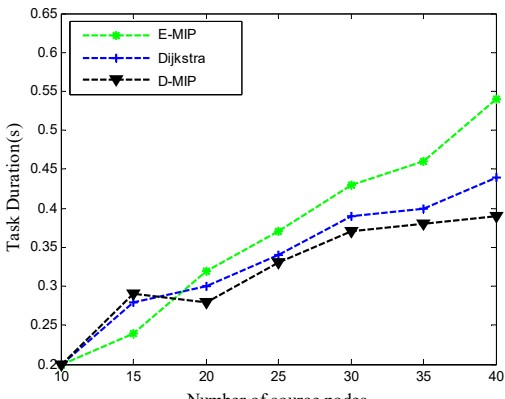

**Figure 12.** The effect of the number of source nodes on energy.

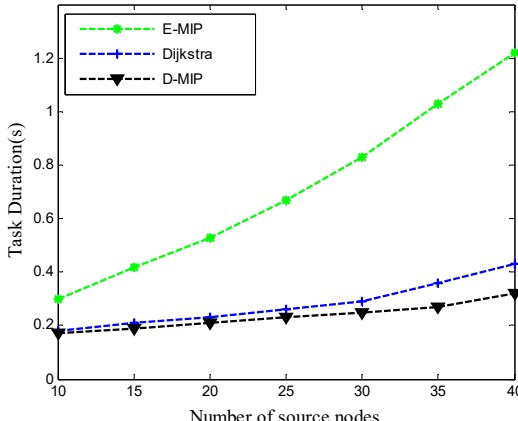

**Figure 13.** The effect of the number of source nodes on task duration.

For time-sensitive WSN applications, energy delay product (EDP) = energy × Delay is defined to evaluate the overall energy consumption and delay performance of the algorithm. The smaller the EDP, the better the performance of the algorithm. The EDP performance of the three algorithms is compared in Figure 14. Because of the worst performance of E-MIP, the EDP is the largest. D-MIP algorithm is 25% lower than Dijkstra algorithm, when n = 40.

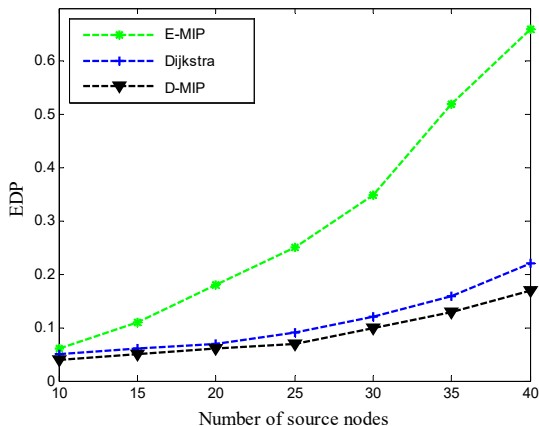

**Figure 14.** The effect of the number of source nodes on EDP.

## 6. Conclusions

Cooperative communication refers to selecting multiple forwarding nodes for cooperative communication when forwarding data, which can improve the reliability of transmission; intelligent mobile agent can collect data in the network, and integrate, compress and encrypt data at the same time, which provides additional flexibility and new functions for the Internet of things network layer. The networks layer cooperative communication and intelligent agent routing algorithm of the IoT are studied in this paper. A new cooperative communication algorithm is proposed. K cooperative nodes are selected for cooperative communication in each hop. The directional source group algorithm based on multi-intelligent agent is studied by using the idea of software definition network. The size of the route is controlled by the angle of the directional sector. In the aspect of intelligent agent, a directional source grouping algorithm D-MIP is proposed, which divides the network into several directional sectors. The source node of the directional sector is assigned to a route. It is shown by a large number of simulations that the D-MIP algorithm can save more energy consumption and time delay than other algorithms. In the future, we will work to solve the problem of dynamic sector angle selection.

In addition, the main experimental results of cooperative communication are based on an independent protocol layer. Many cross-layer theoretical studies, such as cross-layer optimization, have not been proved in experiment. In the aspect of intelligent agent, with the movement of nodes, the data of intelligent agent continuously increases. This causes an increase of forwarding energy consumption of the subsequent source node, which shortens the life of the entire network. It is worthwhile to study the technology of intelligent agent data unloading.

**Author Contributions:** Data curation, Y.Z. (Yongyan Zou) and Y.Z. (Yanzhi Zhang); Investigation, Y.Z. (Yanzhi Zhang); and X.Y.; Methodology, Y.Z. (Yongyan Zou); Project administration, Y.Z. (Yongyan Zou); Software, Y.Z. (Yongyan Zou); and X.Y.

**Funding:** This research received no external funding.

**Conflicts of Interest:** The authors declare no conflict of interest.

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
