# Peer review of "Research on Cooperative Communication Strategy and Intelligent Agent Directional Source Grouping Algorithms for Internet of Things"

_futureinternet, doi:10.3390/fi11110233_

Round 1

Reviewer 1 Report

In this paper, authors study the IoT cooperative communication and intelligent agent source grouping.

The background of grouping nodes is very weak. Authors must improve the state of the art by adding well known papers about this topic such as:

GBP-WAHSN: a group-based protocol for large wireless ad hoc and sensor networks, Journal of Computer Science and Technology 23 (3), 461-480. 2008

And also in mobile nodes such as:

Group-based protocol and mobility model for VANETs to offer internet access, Journal of Network and Computer Applications 36 (3), 1027-1038. 2013

Authors should add their future work at the end of the conclusion section.

References 3 and 26 are the same reference.

Author Response

Comment: In this paper, authors study the IoT cooperative communication and intelligent agent source grouping.

Response: Thanks for your comments. The introduction has been rewritten and some new research has been added.

The background of grouping nodes is very weak. Authors must improve the state of the art by adding well known papers about this topic such as:

GBP-WAHSN: a group-based protocol for large wireless ad hoc and sensor networks, Journal of Computer Science and Technology 23 (3), 461-480. 2008

And also in mobile nodes such as:

Group-based protocol and mobility model for VANETs to offer internet access, Journal of Network and Computer Applications 36 (3), 1027-1038. 2013

Response: Thanks for your comments. The two research results recommended by you have been introduced in the introduction.

Authors should add their future work at the end of the conclusion section.

Thanks for your comments. The future research direction has been introduced in the conclusion.

References 3 and 26 are the same reference.

Response: Thanks for your comments.

Reviewer 2 Report

The paper "Research on Cooperative Communication Strategy and Intelligent Agent Directional Source Grouping Algorithms for Internet of Things" deals with an interesting topic and looks technically sound. However, it is seriously flawed by a poor English style which dramatically hindes the valorization of the proposal. Whereas there are no typos, a deep and full-fledged review by a native speaker is mandatory, in my opinion.
The readability of the paper is also compromised by a non-linear organization of the statements. For example, authors should re-structure the abstract for the sake of readability: research field, issues and motivation, proposal, obtained results. The same holds for the Introduction and Conclusion. Motivations should be better outlined and Bibliography enhanced. Suggested citations: as general reference for agents and IoT; <Gupta, Jayesh K., Maxim Egorov, and Mykel Kochenderfer. "Cooperative multi-agent control using deep reinforcement learning." International Conference on Autonomous Agents and Multiagent Systems. Springer, Cham, 2017.> for cooperative agents; <Qadori, Huthiafa Q., et al. "Multi-mobile agent itinerary planning algorithms for data gathering in wireless sensor networks: A review paper." International Journal of Distributed Sensor Networks 13.1 (2017): 1550147716684841.> for Agent Route Planning in WSN. Edit "Intelligent Agent Algorithms" in "Intelligent Agent" among keywords and "This paper is arranged as follows:" in "This paper is organized as follows:". Insert a comparison table to summarize the outcomes of Section 2 and to highlight the differences between the stateof-the-art and the proposal. Elicit the novelty of the approach. Refer to Sections instead of Chapters in the paper outline. IoT acronym not introduced and full form reported in the conclusion. Avoid algorithm boxes spanning among two pages. Due to all these factors, I believe that before a heavy review of the paper it is not possible, currently, to evaluate the soundness of the proposal. I encourage authors to fulfill the provided comments, improve the readability of the paper and re-submit.

Author Response

Comment: The paper "Research on Cooperative Communication Strategy and Intelligent Agent Directional Source Grouping Algorithms for Internet of Things" deals with an interesting topic and looks technically sound. However, it is seriously flawed by a poor English style which dramatically hindes the valorization of the proposal. Whereas there are no typos, a deep and full-fledged review by a native speaker is mandatory, in my opinion.

Response: Thanks for your comments. The language errors and grammatical errors in this paper have been corrected.

The readability of the paper is also compromised by a non-linear organization of the statements. For example, authors should re-structure the abstract for the sake of readability: research field, issues and motivation, proposal, obtained results. The same holds for the Introduction and Conclusion. Motivations should be better outlined and Bibliography enhanced. Suggested citations: as general reference for agents and IoT; <Gupta, Jayesh K., Maxim Egorov, and Mykel Kochenderfer. "Cooperative multi-agent control using deep reinforcement learning." International Conference on Autonomous Agents and Multiagent Systems. Springer, Cham, 2017.> for cooperative agents; <Qadori, Huthiafa Q., et al. "Multi-mobile agent itinerary planning algorithms for data gathering in wireless sensor networks: A review paper." International Journal of Distributed Sensor Networks 13.1 (2017): 1550147716684841.> for Agent Route Planning in WSN.

Response: Thanks for your comments. The three research results recommended by you have been introduced in the introduction.

Edit "Intelligent Agent Algorithms" in "Intelligent Agent" among keywords and "This paper is arranged as follows:" in "This paper is organized as follows:". Insert a comparison table to summarize the outcomes of Section 2 and to highlight the differences between the stateof-the-art and the proposal. Elicit the novelty of the approach. Refer to Sections instead of Chapters in the paper outline. IoT acronym not introduced and full form reported in the conclusion. Avoid algorithm boxes spanning among two pages. Due to all these factors, I believe that before a heavy review of the paper it is not possible, currently, to evaluate the soundness of the proposal. I encourage authors to fulfill the provided comments, improve the readability of the paper and re-submit.  

Response: Thanks for your comments. These questions have been revised. We have rewritten the introduction and related research, added more writing motivation, and tried to avoid some mistakes.

Round 2

Reviewer 2 Report

Paper has been improved, consider a further English review aiming to a higher readability